# Food Security and Cardio-Metabolic Risk in Individuals with Metabolic Syndrome [note 1]

**DOI:** 10.3390/ijerph22010028

**Published:** 2024-12-29

**Authors:** Bong Nguyen, Barbara Lohse, Lynda H. Powell, Kevin S. Masters, Jannette Berkley-Patton, Betty M. Drees

**Affiliations:** 1Department of Biomedical and Health Informatics, University of Missouri, Kansas City, MO 64108, USA; berkleypattonj@umkc.edu (J.B.-P.); dreesb@umkc.edu (B.M.D.); 2Wegmans School of Health and Nutrition, Rochester Institute of Technology, Rochester, NY 14623, USA; balihst@rit.edu; 3Department of Family and Preventive Medicine, Rush University Medical Center, Chicago, IL 60612, USA; lynda_powell@rush.edu; 4Anschutz Medical Campus, University of Colorado Denver, Denver, CO 80045, USA; kevin.masters@ucdenver.edu

**Keywords:** cardiometabolic risk, diabetes, food insecurity, metabolic syndrome

## Abstract

This study assessed the association of food security with potential cardio-metabolic risk factors among persons with metabolic syndrome (MetS). Data were derived from the baseline data of a randomized controlled lifestyle intervention trial for individuals with MetS. Household food security, fruit and vegetable intake, perceived food environment, and perceived stress were collected using validated questionnaires. Cardio-metabolic measures assessed with standardized procedures included body mass index, waist circumference, blood pressure, glucose, HbA1c, and lipids. Regression models adjusted for demographics, medication use, and perceived stress were performed. Of a total of 664 participants (median age 56), the majority were female, non-Hispanic White, college-educated, and employed. Food insecurity affected 23% (n = 152), with 5% (n = 31) experiencing very low food security. Food-insecure individuals had significantly higher stress (*p* < 0.001), lacked healthy food access (*p* < 0.001), were and less likely to consume ≥2 servings of vegetables/day (*p* = 0.003). HbA1c was the only cardio-metabolic measure significantly associated with food security (*p* = 0.007). The link between food insecurity and elevated HbA1c levels highlights the importance of addressing food insecurity and stress to improve metabolic health outcomes in the MetS population.

## 1. Introduction

Metabolic syndrome (MetS) is a group of multiple metabolic abnormalities associated with cardiovascular disease (CVD) that occur together, including increased blood pressure, hyperglycemia, excess body fat around the waist, and abnormal cholesterol and triglyceride levels [1]. An individual with three or more of these conditions is deemed to have MetS [2]. This is an ongoing problem in the US, with the prevalence increasing from 37.6% in 2011–2012 to 41.8% in 2017–2018 [3]. Individuals with MetS have a significantly higher risk for several adverse health outcomes compared to those without MetS. For instance, findings from a meta-analysis indicated that individuals with MetS had a 22% higher risk of dying from any cause, a 36% higher risk of dying from CVD, and a nearly 50% higher risk of having a stroke [4]. The odds of developing coronary heart disease and heart attack were over 75% higher in individuals with metabolic disorders compared to those without MetS (*p* < 0.001) [5]. In addition, individuals with MetS face increased economic challenges, with greater medication expenditures, frequent hospitalizations, and higher utilization of outpatient and physician services [6,7]. This economic burden could result in fewer financial resources available for food [8].

Food insecurity, defined as a lack of regular access to enough safe and nutritious food for normal growth and development and an active and healthy life [9], is a public nutrition concern in both high- and low-income countries, including the United States, which has a rate of 12.8% in 2022 [10]. It is associated with a higher risk of MetS [11,12]. In a recent analysis report using NHANES 2005–2016, Reeder and Reneker found that individuals with food insecurity, particularly women, are more likely to have MetS, especially those with very low food security [13]. Food insecurity has been found to be associated with a poor-quality diet and disrupted eating patterns [14,15]; however, recent meta-analysis suggested that dietary factors did not mediate the association between food insecurity and metabolic health, as previously thought [16]. Other factors, including stress and other social determinants of health, might play a role in the association.

Given the dual burden of food insecurity and MetS, this study aimed to fill critical gaps by examining the relationship between food security status and key cardiometabolic risk factors. Specifically, the study explored the associations between food insecurity and cardiometabolic risk factors. Identifying these associations is crucial for developing targeted interventions to alleviate the adverse health outcomes experienced by this vulnerable population.

## 2. Materials and Methods

### 2.1. Study Design and Sample

This cross-sectional study used the baseline data collected from October 2019 to February 2022 in the 5-site Enhanced Lifestyles in the Metabolic Syndrome (ELM) trial of individuals with MetS (ClinicalTrials.gov, NCT04036006). The study sample was recruited using various strategies, such as electronic medical records, medical provider referrals, and self-referral, which was described previously [17,18]. Eligible respondents included adults who had a diagnosis of MetS defined as having at least 3 of 5 indicators: (1) waist circumference ≥102 cm (40.2 inches) in men or ≥88 cm (34.6 inches) in women; (2) systolic blood pressure ≥130 mm Hg or diastolic blood pressure ≥85 mm Hg, or use of blood pressure medications; (3) fasting blood glucose 100–125 mg/dL or use of metformin; (4) triglycerides ≥150 mg/dL or treatment of hypertriglyceridemia; or (5) HDL cholesterol <40 mg/dL for men or <50 mg/dL for women, or treatment of low HDL cholesterol. Exclusions were based upon safety concerns, logistical barriers that would compromise internal validity, and treatments that would confound trial outcomes. A complete list of the inclusion and exclusion criteria was published in the ELM design paper [17].

A total of 664 respondents were confirmed to have MetS and completed all components of baseline assessment; among these individuals, 618 were randomized into the ELM Trial. In this study, analyses included all 664 respondents irrespective of randomization. The study was approved by the Rush University Institutional Review Board (as a central review board for all 5 sites) and informed consent was obtained before any data were collected.

### 2.2. Data Collection

Research assistants at each study site were trained to administer self-reported surveys and to collect physical measures using standardized protocols [17]. They read all survey questions to the respondents and filled in the answers to limit missing data.

#### 2.2.1. Anthropometric and Bioclinical Data Collection

Research assistants measured blood pressure (BP), waist circumference (WC), height, and weight using standard protocols [17]. WC was measured with a Seca 201 tape at the top of hip bone (iliac crest) in the standing position, arms crossed and on opposite shoulders. Height was measured with a Seca 213 stadiometer to the nearest 0.1 cm and weight was measured with a Seca 876 flat scale and recorded to the nearest 0.1 kg. WC, height, and weight were measured 2 times and then the average of the 2 measurements was calculated. Body mass index (BMI) was calculated as weight divided by the square of height (kg/m^2^).

Blood pressure was measured with an Omron^®^ HEM-907XL automatic pressure monitor following a standard procedure. Participants rested 5 min before the first reading was taken. A total of 3 readings were measured for BP and the average of 3 readings was recorded.

The 12 h fasting blood collection was performed by trained phlebotomists and sent to Quest Diagnosis for analysis of plasma glucose, glycated hemoglobin A1c (HbA1c), and lipid panel (triglycerides, total cholesterol, high-density lipoprotein (HDL) cholesterol, and low-density lipoprotein (LDL) cholesterol).

#### 2.2.2. Self-Reported Instruments

Food security: food security was measured using the validated USDA 10-item survey to examine the ability of a household to meet basic food needs over the past 12 months [19]. Households were classified as food secure if they reported no food-insecure conditions or if they reported only one or two food-insecure conditions (scores 0–2). Households were coded as food insecure if they reported 3 or more food-insecure conditions (scores 3–10), and then further classified as having either low food security (scores 3–5) or very low food security (scores of 6–10).

Fruit and vegetable (FV) intake: The National Cancer Institute fruit and vegetable intake screener was used to estimate daily number of servings of FV over the past month [20]. The questionnaire included intake of 100% fruit juice, fruits, and 8 vegetable groups (lettuce salad, French fries or fried potatoes, other white potatoes, cooked dried beans, other vegetables, tomato sauce, vegetable soup, and mixtures that included vegetables). Responses on average frequency of FV consumption over the past month were converted to total daily servings of FV intake. The questionnaire also had subscales for vegetable intake and fruit intake.

Perceived neighborhood food environments: Perceived availability and quality of fresh FV and low-fat products within the neighborhood were measured using 3-statement items that were previously validated and tested for reliability [21,22,23]. Each item provided the following response options: “strongly disagree” (1), “disagree” (2), “neither disagree or agree” (3), “agree” (4), and “strongly agree” (5). Responses were dichotomized into “did not agree” (≤3) and “agreed” (≥4). Cronbach’s α = 0.9.

Perceived stress: The Cohen Perceived Stress Scale was used to measure the degree to which individuals perceive situations in their lives as stressful [24]. The scale consisted of 14 questions to assess the frequency and intensity of perceived stress over the past month. Respondents rated each item on a 5-point Likert scale, ranging from “never” to “almost never”, “sometimes”, “fairly often”, or “very often”, with scores of 0 to 4, respectively. Seven items were reverse scored, and the responses were then summed for a total score that could range from 0 to 56, with higher scores indicating greater perceived stress. In addition, scores were categorized as low stress (0–18), moderate stress (19–37), and high stress (38–56). Cronbach’s alpha coefficient in this study’s sample was 0.8, suggesting internal consistency of the scale was reliable.

### 2.3. Statistical Analysis

Statistical analyses were performed with SPSS Statistics version 29. Normality of data was examined using the Shapiro–Wilk test and data with non-normality were log-transformed before analysis. Triglycerides and glucose levels were normally distributed after a log base 10 transformation. Differences in respondent characteristics by three food security groups (food security, low food security, and very low food security) were assessed using chi-square tests for categorical variables (gender, race, education, relationship, employment, find it difficult to pay for basics, BMI categories, and levels of perceived stress) and one-way ANOVA for continuous variables (age, household income, servings of FV intake, perceived stress score, BMI, systolic and diastolic BP, triglyceride, total and HDL cholesterol, fasting glucose, and HbA1c levels). The association of food security status (food security coded as 0 versus food insecurity coded as 1) with the cardiometabolic dependent variables (fasting glucose, HbA1c, total cholesterol, HDL cholesterol, triglycerides, systolic and diastolic BP) was examined using a series of linear regression models, unadjusted and adjusted for covariates including demographic characteristics (age, race, relationship status, employment, household income), medication use, BMI, and perceived stress. Statistical significance was set at *p* < 0.05.

## 3. Results

### 3.1. Respondent Characteristics

The study respondents’ ages ranged from 18 to 84 years, with a median of 56. The majority were female, non-Hispanic White, college-educated, and employed either full-time or part-time. Food insecurity was present in 152 respondents (23%), including 18.2% (n = 121) who had low food security and 5% (n = 31) who had very low food security. Out of the 664 respondents, 80 (12%) refused to report or did not know their household income in the previous year; however, these individuals did not significantly differ in demographic characteristics and food security status (except employment) from those who reported their household income. There was a significant graded relationship between progressive levels of food insecurity and lower income: respondents with very low food security had a mean household income of $46,018 ± 31,222, significantly higher compared to low food security groups ($83,444 ± 67,310) and food secure individuals ($109,517 ± 76,795) (*p* < 0.001). Paying for the basics, such as food, housing, medical care, and heating, significantly differed by food security status, as the majority of participants with very low food security (83.9%) and slightly over a fourth of those with low food security (26.4%) reported having some difficulty paying for basics, while the same difficulty was reported by 7.2% of individuals with food security (*p* < 0.001). Metformin use was most prevalent among participants with very low food security (22.6%), compared to 13.2% of those with low food security and 9.2% of food-secure individuals (*p* = 0.036). Detailed information is shown in Table 1.

Mean BMI was 36.5 ± 7.01 kg/m^2^, with 1% deemed normal (BMI 18.5 to <25), 16% overweight (BMI 25 to <30), and 83% obese (BMI is ≥30). There were no significant differences in BMI or its category by food security status.

The mean score of perceived stress among individuals with very low food security was 24.6 ± 6.6, significantly higher than the score observed in individuals with food security and low food security (19.3 ± 6.6 and 21.6 ± 6.3 resp.; *p* < 0.001). In addition, a greater proportion of individuals with very low food security (41.9%) reported experiencing high stress compared to their counterparts with food security and low food security (*p* < 0.001).

Only HbA1c levels were significantly different by food security group (*p* = 0.012); specifically, participants with very low food security had a significantly higher level of HbA1c (5.8 ± 0.3) compared to those with low food security (5.7 ± 0.4) and food security (5.6 ± 0.4). Fasting glucose levels were significantly associated with food security status in unadjusted models (*p* = 0.011); however, this significance did not remain (*p* > 0.05) after adjusting for the covariates of (Table 1). Total cholesterol levels, HDL cholesterol, triglyceride, and blood pressure were not significantly different between food security groups in either adjusted or unadjusted models.

### 3.2. Dietary Quality

Both fruit and vegetable intakes were assessed in this study; however, significant differences were found in vegetable intake across food security groups. Consuming at least two servings of vegetables a day was associated with being food secure (*p* = 0.003).

Table 2 shows that the perceptions of the availability and quality of fresh FV and low-fat products in the neighborhood significantly differed by food security status (*p* < 0.001). More individuals with low and very low food security disagreed with there being a large selection of fresh FV and doubted the quality in their neighborhood compared to those with food security. Almost 15% to 20% of the individuals with low and very low food security disagreed about the availability of a large selection of low-fat products within their neighborhood, while 5.7% of those with food security had a similar disagreement.

### 3.3. Association Between Food Security Status and Level of HbA1c

Table 3 shows the association of food insecurity with HbA1c levels, adjusted for demographic characteristics (age, race, relationship status, education, household income, and employment), medication use, perceived stress, and BMI. The results indicated that food insecurity was positively associated with HbA1c levels, with individuals experiencing food insecurity having HbA1c levels 0.1 higher than those with food security (*p* = 0.007).

In addition to food security, other factors were found to be independently associated with HbA1c levels, including perceived stress, age, race, and medication use.

## 4. Discussion

This study identified a prevalence of food insecurity among individuals with MetS that was almost double the rate of 12.8% among the general population [10]. Higher rates of food insecurity have been reported among individuals with cardiometabolic conditions such as hypertension, diabetes mellitus, and CVD [25,26,27,28,29]. Leung et al. found in their study that nearly a third of the participants (31.7%) with diabetes experienced food insecurity, while almost half of those with high blood pressure (46.7%) were food insecure [30]. Findings from a review by Kirby, Bernard, and Liang showed that the prevalence of food insecurity among adults with diabetes was almost double the rate of those without diabetes (16% vs. 9%) [31]. These findings highlight the disproportionately high prevalence of food insecurity among individuals with MetS and other cardiometabolic conditions, underscoring the need for targeted interventions to address both food insecurity and chronic disease management in the populations.

The findings of this study suggest that food insecurity, especially at more severe levels, may contribute to poorer glycemic control. This relationship persisted even after adjusting for other potential confounders, further supporting the robust link between food insecurity and higher HbA1c. While a 0.2% difference in HbA1c may seem small, it can have substantial implications for long-term glycemic control and the prevention of diabetes-related complications. The results align with prior research suggesting that food insecurity can negatively impact health outcomes, particularly among individuals with diabetes or those at risk of developing it [32,33].

Food insecurity creates a pervasive sense of uncertainty and anxiety about food availability, which contributes to high levels of stress, as consistently demonstrated in research studies [34,35,36]. In a recent report of the 3-day workshop “Food insecurity, Neighborhood Food Environment, and Nutrition Health Disparities: State of the Science”, organized by the National Institutes of Health in collaboration with the Centers for Disease Control and Prevention (CDC) and the United States Department of Agriculture (USDA) in September 2021 [37], Odoms-Young et al. reported findings from a systematic review and meta-analysis involving over 300,000 people across 10 countries. The review revealed that food insecurity was linked to 40% higher odds of depression and 34% higher odds of stress universally [38]. This current study among the MetS population further contributes to the existing body of evidence. The sustained high levels of perceived stress among food-insecure individuals may disrupt metabolic processes, leading to dysregulation of glucose metabolism, insulin resistance, and inflammation—all hallmark features of MetS [39,40,41]—thus further compounding their risk of poor glycemic control. The link between food insecurity, stress, and metabolic dysregulation underscores the importance of incorporating food security interventions into comprehensive care for individuals with MetS. Addressing food insecurity may not only alleviate stress but also help mitigate the negative metabolic impacts, ultimately improving health outcomes in this population. Social determinants of health, such as age and race, must also be considered in intervention design.

A strength of this study is its inclusion of a large sample size and geographically diverse participants with a confirmed diagnosis of MetS, which provided adequate statistical power for the analyses. Although some of the data (i.e., food security status, food availability, and FV intake) were self-reported, these survey items have been validated. The use of measurement techniques following an interview protocol to collect self-reported data limited missing data. With respect to limitations, the sample was predominantly middle-aged women, indicating that the data may not be generalizable to other MetS populations. The emergence of the COVID-19 pandemic caused significant disruption in people’s lives, such as increased unemployment and reduced household income, fresh produce and supply chain shortages, and increased food prices due to inflation [42,43]. Study data were collected before and during the COVID-19 pandemic occurred, which might have impacted the participants’ fruit and vegetable intakes and their responses to the food availability and food security surveys.

## 5. Conclusions

The findings revealed a high prevalence of food insecurity among individuals with metabolic syndrome, with those experiencing food insecurity reporting a lack of access to healthy and quality foods, a decreased likelihood of consuming recommended servings of vegetables, and high perceived stress. Moreover, the study identified a graded relationship between food insecurity and HbA1c levels in individuals with MetS, suggesting potential implications for metabolic health management. These findings underscore the need for comprehensive interventions and policies targeting food insecurity and stress management to alleviate adverse health outcomes in this vulnerable population.

## Figures and Tables

**Table 1 ijerph-22-00028-t001:** Socio-demographic characteristics of participants with confirmed metabolic syndrome.

Characteristics	Total Sample (n = 664)	Food Security (n = 512)	Low Food Security (n = 121)	Very Low Food Security (n = 31)	*p*-Value
Female, n (%)	501 (75.5)	386 (75.4)	92 (76.0)	23 (74.2)	0.976
Age (years), median; range	56; 18–84	57; 18–84	52; 28–74	50; 20–72	<0.001 ***
Hispanic, n (%)	66 (9.9)	41 (8.0)	22 (18.2)	3 (9.7)	0.003 **
Race ^a^, n (%)					<0.001 ***
White	489 (73.6)	400 (78.1)	73 (60.3)	16 (51.6)	
Black	115 (17.3)	68 (13.3)	33 (27.3)	14 (45.2)	
Multi/Others ^b^	60 (9.0)	44 (8.6)	15 (12.4)	1 (3.2)	
Education ^c^, n (%)					0.003 **
High school or less	76 (11.4)	54 (10.5)	12 (10.7)	9 (29.0)	
GED, vocational school, some college/associate’s degree	195 (29.4)	141 (27.5)	40 (33.1)	14 (45.2)	
Bachelor’s or higher degree	389 (58.6)	313 (61.1)	68 (56.2)	8 (25.8)	
Others	4 (0.6)	4 (0.8)	0	0	
Relationship status, n (%)					<0.001 ***
Single	148 (22.3)	95 (18.6)	42 (34.7)	11 (35.5)	
Living with partner/spouse	410 (61.7)	342 (66.8)	57 (47.1)	11 (35.5)	
Divorced/living separated or widowed	106 (15.9)	75 (14.6)	22 (18.1)	9 (29.1)	
Employment status, n (%)					0.002 **
Working full-time or part-time	468 (70.5)	357 (69.7)	87 (71.9)	24 (77.4)	
Retired	138 (20.8)	120 (23.4)	14 (11.6)	4 (12.9)	
Not working/disabled/ill	58 (8.8)	35 (6.8)	20 (16.5)	3 (9.7)	
Household income ($), mean ± SD	$101,843 ± 75,236	$109,517 ± 76,795	$83,444 ± 67,310	$46,018 ± 31,222	<0.001 ***
Find it difficult to pay for basis, n (%)					<0.001 ***
Very or somewhat hard	95 (14.3)	37 (7.2)	32 (26.4)	26 (83.9)	
Not hard at all	560 (84.3)	472 (92.2)	84 (69.4)	4 (12.9)	
Perceived stress score, mean ± SD	20.0 ± 6.7	19.3 ± 6.6	21.6 ± 6.3	24.6 ± 6.6	<0.001 ***
Perceived stress level, n (%)					<0.001 ***
Low and moderate stress	561 (84.5)	443 (86.6)	100 (82.7)	18 (58.1)	
High stress	103 (15.5)	69 (13.5)	21 (17.4)	13 (41.9)	
Use of medicines, n (%)					
Lipid	26 (3.9)	26 (5.1)	0	0	0.022 *
Blood pressure	432 (65.1)	340 (66.4)	72 (59.5)	20 (64.5)	0.358
Metformin	70 (10.5)	47 (9.2)	16 (13.2)	7 (22.6)	0.035 *
Dietary quality, n (%)					
≥2 servings of vegetable/day	405 (60.8)	328 (64.1)	59 (48.8)	18 (58.1)	0.003 **
≥3 servings of vegetable/day	246 (36.9)	199 (38.9)	35 (28.9)	12 (38.7)	0.075
BMI, mean ± SD	36.5 ± 7.0	36.2 ± 6.9	37.1 ± 7.1	38.5 ± 8.7	0.104
BMI category, n (%)					0.079
Normal/healthy weight	7 (1.1)	6 (1.2)	1 (0.8)	0	
Overweight	108 (16.2)	94 (18.4)	13 (10.7)	1 (3.2)	
Obesity	549 (82.7)	412 (80.5)	107 (88.4)	30 (96.8)	
Fasting glucose ^d^, mean ± SD	100.2 ± 12.8	98.3 ± 12.9	98.3 ± 13.5	95 ± 12.9	0.011 *
HbA1c	5.65 ± 0.4	5.6 ± 0.4	5.7 ± 0.4	5.8 ± 0.3	0.012 *
Total Cholesterol	192.8 ± 42.0	189.5 ± 41.1	190 ± 37.0	184.8 ± 38.3	0.203
HDL Cholesterol	47.0 ± 11.0	45.5 ± 10.8	47.2 ± 12.6	42.9 ± 8.1	0.415
Triglycerides	174.9 ± 95.4	164.0 ± 76.1	152.0 ± 62.9	153.1 ± 70.9	0.311
Systolic BP	128.5 ± 15.6	128 ± 15.8	127 ± 14.6	128.1 ± 15.1	0.694
Diastolic BP	83.3 ± 10.1	83.2 ± 10.2	85 ± 9.2	83.7 ± 10.6	0.494

^a^ Analyses were performed to compare differences between Black and White. ^b^ Other races include Asian, American Indian/Alaskan Native, Native Hawaiian, or Other Pacific Islander. ^c^ Differences were assessed between three education statuses: high school or less; GED, vocational school, some college/associate’s degree; and bachelor’s or higher degree. ^d^
*p*-values were not significant after controlling for demographic characteristics (age, race, relationship, education, household income, employment), metformin use, BMI, and perceived stress. * *p* < 0.05; ** *p* < 0.01, *** *p* < 0.001. BMI, body mass index; BP, blood pressure; GED, general educational diploma; HDL, high-density lipoproteins; SD, standard deviation.

**Table 2 ijerph-22-00028-t002:** Perception on the availability and quality of foods within the neighborhood (agreement only).

	Total Sample	Food Security	Low Food Security	Very Low Food Security	*p*-Value
High quality of fresh FVs in neighborhood	609 (91.7)	485 (94.7)	103 (85.1)	21 (67.8)	<0.001
Large selection of fresh FVs available in neighborhood	628 (94.6)	496 (96.9)	108 (89.2)	24 (77.5)	<0.001
Large selection of low-fat products available in neighborhood	611 (92.0)	483 (94.3)	103 (85.2)	25 (80.7)	<0.001

**Table 3 ijerph-22-00028-t003:** Association of food security status and HbA1c.

Independent Variables	Unadjusted Coefficient	Unadjusted 95% CI	*p*-Value	Adjusted Coefficient ^a^	Adjusted 95% CI	*p*-Value
Food security status ^b^	0.09	0.02; 0.15	0.012	0.10	0.028; 0.18	0.007
Perceived stress	−0.01	−0.02; 0.00	0.006	−0.01	−0.02; 0.00	0.007
Age	0.004	0.001; 0.01	0.003	0.01	0.00; 0.01	0.001
Race ^c^	−0.25	−0.33; −0.18	<0.001	−0.22	−0.30; −0.14	0.008
Medication use ^d^	0.19	0.10; 0.28	<0.001	0.18	0.08; 0.28	<0.001

^a^ adjusted model with covariates of demographic characteristics (age, race, relationship status, education, household income, employment), medication use, BMI, and perceived stress. However, relationship status, education, household income, employment, and BMI were not significant in the adjusted model; therefore, these characteristics were excluded from the table. ^b^ reference group, food security; ^c^ reference group, Black/African American; ^d^ reference group, not using medications.

## Data Availability

Research data are available upon request to the corresponding author and L.H.P., the Principal Investigator of this research.

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
