# Peer review of "Food Security and Cardio-Metabolic Risk in Individuals with Metabolic Syndromeâ€"

_ijerph, 2024, doi:10.3390/ijerph22010028_

Round 1
Reviewer 1 Report
Comments and Suggestions for Authors
I have reviewed the paper tilted ‘Food security and cardio-metabolic risk in individuals with metabolic syndrome’. This study aimed to assess association of food security with cardio-metabolic risk factors in persons with metabolic syndrome (MetS).
I have some suggestions, as follows:
Major concerns
Introduction
1. This section should be complemented by up-to-date study material and refer to the overview of the association between metabolic syndrome and food insecurity. I suggest to review the meta-analyses, too. For example, some Authors reported that dietary factors are not the main factors underlying the association of food security with metabolic health.
Materials and methods
2. Line 56: The Authors wrote: ‘Study sample and recruitment have been previously described’ [15,16]. Suggestion: References correspond to 2022-24; however, this study was carried out between 2019 and 2022. It seems necessary to clarify this information and clearly describe, as follows:
(a) What was the design of this study?
(b) How was the representative size of the study sample calculated?
(c) Which sampling technique was applied?
(d) What exclusion criteria were considered and applied in order to form the sample studied?
3. Lines 111-113: The Authors wrote: ‘Perceived stress: The Cohen Perceived Stress Scale was used to measure the degree to which individuals perceive situations in their lives as stressful. The scale consisted of 14 questions to assess the frequency and intensity of perceived stress over the past month’. Suggestion: The Perceived Stress Scale (PSS) is a classic stress assessment instrument. The tool, while originally developed in 1983, remains a popular choice for helping us understand how different situations affect our feelings and perceived stress. However, this, also known as PSS-10, has 10 but not 14 questions. Thus, how could the Authors explain these methodological inconsistencies? Has a validated study instrument was used to assess perceived chronic stress in subjects?
4. What freeware/software was used to perform statistical data analysis?
Results and Discussion
5. Lines 196-197: The Authors wrote: ‘The nature of this study limits understanding of the directionality of the association, but it might be likely bi-directional…’. Suggestion: Exclusively in the methodological section, The Authors must consider the dependent and independent variables very clearly. A statistical data analysis should be performed in accordance with the specific design of the study. Also, the regression model is designed to make a prognosis on how the independent variables can predict the dependent variable.
6. In addition, Table 3 lacks association between perceived chronic stress and food insecurity. Add this information, please.
7. Also, I think that perceived chronic stress can not be associated with food insecurity. I suggest that PSS-10 scale can serve only as confounder in this study.
8. Finally, no evidence from this study can not be applied to Clinical Practice, as the Authors conducted a single cross-sectional study with (non) representative sample size and non-probabilistic sampling technique. For this case, there is a need for longitudinal studies. I think that subsection, namely, ‘Clinical Recommendations’ should be declined.
9. All in all, the paper needs much work before acceptation, as severe inconsistencies were observed throughout the entire manuscript.
Minor concerns
10. References should be adjusted to recommendations of MDPI in-house or ACS style.
11. Institutional Review Board Statement: Add the number and date when issue was approved.
12. Keywords must be displayed in alphabetical order.
13. Line 2: The term ‘cardio-metabolic risk’ should be changed to ‘potential cardio-metabolic risk’.
Author Response
Dear Reviewer,
Thank you very much for your thorough review and providing insight comments to help us improve our paper. Please find the attachment for our responses to your comments.
Sincerely!
Bong Nguyen

Reviewer 2 Report
Comments and Suggestions for Authors
This is an interesting cross-sectional study investigating the association of food security with cardio-metabolic risk factors among persons with metabolic syndrome. The study design is epidemiologically oriented, and the presentation of its research objectives (as well as the data collection) is done in an analytical and precise manner.
I would like to ask a specific question and make some suggestions:
Α) In the data collection and in all analyses, the assessment of the participants' physical activity is completely absent, which is a disadvantage, given the relationship that physical exercise has with the parameters of metabolic syndrome. I would like to ask the authors why they did not assess the participants' physical activity with a questionnaire (while they have an assessment of stress and other factors) and how they would answer questions regarding the possible confounding effect of physical exercise on the results of the study.
Β) In Τable 1, the percentages shown in parentheses should be calculated for each row of the table (i.e., for each socio-demographic characteristic) and not for the categories of the Food Insecurity variable. For example, in the race variable, for Whites the percentages should be 400/489=81.8% in the Food security category, 73/489=14.9% in the Low food security category, and 16/489=3.3% in the Very low food security category. The same should be done for the remaining socio-demographic characteristics of the table.
C) The way the regression results are presented in Table 3 is paradoxical and incomprehensible. It should be mentioned somewhere in the text that the response variables of the analyses are the cardiometabolic risk factors, i.e. fasting glucose, HbA1c, Total Cholesterol etc. It should also be defined which category of the Food insecure variable is the reference category of the analyses, while for the remaining categories (of Food insecure variable) the corresponding regression coefficients (b) with their confidence intervals should be reported in the table.
D) In line 152 of the manuscript the phrase very food food secure should become very low food secure
Author Response
Dear Reviewer,
Thank you very much for reviewing our manuscript and providing comments to help us improve our paper. Please find the attachment for our responses to your comments.
Sincerely,
Bong Nguyen

Reviewer 3 Report
Comments and Suggestions for Authors
The manuscript is of interest, but needs to be improved in some aspects to be publishable:
The abstract must contain an introduction and indicate the type of study. It must indicate that HbA1c was measured and is only indicated in the results section.
The bibliographic references in the manuscript are not adapted to the journal format. Authors must read the instructions for authors and thoroughly review the manuscript. They are indicated with superscript when they should be in brackets. The references section is also not correct.
The introduction is too short and should reflect what has been done so far and why the research was carried out.
The methodology needs to be improved in the following aspects:
- Indicate the type of study.
- Indicate whether they meet the criteria for the type of study. I deduce that this is an observational and cross-sectional epidemiological study, so the STROBE Checklists should be reviewed: https://www.strobe-statement.org/checklists/
- A reference is required on lines 57-63 regarding the definition of MetS.
- The bibliographic references for “Research assistants measured blood pressure (BP), waist circumference (WC), height, 75 and weight using standard protocols” should be mentioned.
- The correct name for the anthropometric measurements is:
o Waist girth instead of waist circumference.
o Stretch Stature instead of height.
o Body mass instead of weight.
- Authors should check whether the waist circumference included in the MetS refers to the minimum or maximum waist (hip bone (iliac crest))
In the discussion section, they should compare with other studies, indicating specific data or similarities with the rest of the research, without making a general allusion.
In the discussion section, a section on the limitations of the study (last paragraph of the discussion) should be included, as well as its strengths.
Author Response

(The authors gave the same response as above.)

Round 2
Reviewer 1 Report
Comments and Suggestions for Authors
Good job!
Author Response
Dear Reviewer,
Thank you very much for your review to help improve the paper!
Reviewer 2 Report
Comments and Suggestions for Authors
Regarding the authors' responses, I have the following to point out:
The assessment of physical activity should have been foreseen in some way during the design of the study. The authors' response that they could not calculate it according to the 10-day turnaround requirement of submitting revision does not stand. PA is a key moderator in the estimation of HbA1c level and cannot be ignored. Without it, the regression models that are run are meaningless and the estimators that result are biased. There are ways to put some indication of physical activity in multivariate analyses. The simplest one that comes to mind is the energy intake of the participants.
The presentation and interpretation of regression models is still problematic. Authors should put the regression coefficients in their tables and not betas, which are nothing more than the partial correlation coefficients of each covariate with the response variable (after controlling for the linear effects of the remaining covariates).
The authors, when using and interpreting the regression coefficients, must take care of the following:
1) The Food Security Status variable is incorrectly entered into the models without being recoded using dummy variables. From what I understand, it is entered into the models as a continuous variable. This creates incorrect estimates of the regression coefficients.
2) For each categorical predictor variable of the models, the reference category must be defined and, therefore, the resulting regression coefficients will define the adjusted differences of the other categories with respect to the reference one (this also concerns the Food Security Status variable).
3) The interpretation of the effects of each predictor variable on HbA1c level should be done through the regression coefficients.
4) The mean values ​​that the authors compare in the results (e.g. for the food security status variable: 5.8 ± 0.3, 5.7± 0.4 and 5.6± 0.4, page 7) are the unadjusted mean values ​​without controlling for the remaining covariates. The authors should make comparisons based on the adjusted values ​​resulting from the regression.
They may need to consult a biostatistician to help them run the regression models.
Author Response
Dear Reviewer,
Thank you very much for your thorough review and suggestion to improve our paper.
We collected physical activity (PA) levels as average weekly minutes of PA and average daily steps by having participants wear an accelerometer for 7 consecutive days. We run analysis of the association of PA levels with HbA1c or food security status; however there were no significant associations in the study sample. Therefore we did not include PA data in this paper. Please check the outputs attached.
Thank you for pointing out the regression information and table presentation. We updated our Table 3 to present regression coefficients instead of beta coefficients and added information regarding reference group for categorical variables. Please check our updated Table 3, data analysis and results section (Page 7).
Thank you very much!

Reviewer 3 Report
Comments and Suggestions for Authors
The authors have incorporated the information requested by the reviewers, so it can be accepted for publication.
Author Response
Dear Reviewer,
Thank you very much for your review to help improve our paper!